# Process Development Methods in Microtechnology and the Associated Process Environment

**DOI:** 10.3390/mi16060606

**Published:** 2025-05-22

**Authors:** Korbinian T. Metz, Faruk Civelek, André Zimmermann

**Affiliations:** 1Faculty 7, Institute for Micro Integration (IFM), University of Stuttgart, 70569 Stuttgart, Germany; andre.zimmermann@ifm.uni-stuttgart.de; 2Realien GmbH, 72666 Neckartailfingen, Germany; 3Industrial Engineering, Baden-Wuerttemberg Cooperative State University (DHBW), 70174 Heidenheim, Germany; faruk.civelek@dhbw-heidenheim.de; 4Hahn-Schickard-Gesellschaft für Angewandte Forschung e.V., 70569 Stuttgart, Germany

**Keywords:** MEMS, microsystems technology, process development, manufacturing process chains, process environment

## Abstract

Microsystem technology (MST) and micro-electro-mechanical systems (MEMS) are key technologies that continually introduce new application opportunities. Increasing complexity and individualization require systematic process development to avoid errors and delays. While existing methods for process development address various aspects of the manufacturing process, the systematic consideration of external factors influencing the process environment (PEnv) remains broadly inadequate. Despite extensive standards, PEnv-related influences lead to quality fluctuations in practice. A list of influencing factors and an example process illustrate these challenges. This study aims to analyze which methods exist for process development in MST and to what extent they systematically consider process environmental factors. A mixed methods design was used for the analysis. In a systematic literature review (SLR) using traditional databases and Artificial intelligence-supported search tools, a total of 75 relevant studies from the years 2005 to 2024 were identified. The methods that cover various aspects of process development are presented in an overview. An adapted GRADE (Grading of Recommendations Assessment, Development, and Evaluation) analysis was used to check the extent to which the PEnv can be included in process development using the methods currently available. The results show that existing approaches often take PEnv into account insufficiently. Efficient consideration with the use of current methods requires extensive expert knowledge, knowledge management, and project-specific supplementary methods. This study emphasizes the need for research into methods that systematically integrate environmental requirements into process development to improve the efficiency and quality of MST manufacturing in this area.

## 1. Introduction

Microsystems technology (MST) and especially micro-electro-mechanical systems (MEMS) are of outstanding importance as key technologies of our time [1,2,3,4]. Their continuous development and increasing importance continuously open up new possibilities and applications, which are being intensively researched [2,5,6,7]. Development errors in products and manufacturing processes lead to considerable costs and significant production delays [8,9]. Errors resulting from insufficient or late identification of requirements underline the need for an early and structured approach to process and product development [10,11]. Fast and systematic development is essential to react efficiently to dynamic market requirements. This applies particularly to MST, where the trend is toward individualized products [11]. MST’s high degree of individuality and complexity is reflected in the so-called MEMS law, which states that almost every product requires an individual manufacturing process (one product–one process) [12,13,14,15,16].

The ongoing miniaturization of structures is a key driver of this increasing complexity. With the progression of More Moore—the continuous downscaling and densification of transistors according to Moore’s Law—and the increasing relevance of More than Moore, where additional functionalities such as sensors or MEMS are heterogeneously integrated, both the complexity and sensitivity of manufacturing processes in MST are significantly increasing [17]. As a result, even minimal deviations in process environment (PEnv) parameters can compromise component quality. A stable and well-controlled PEnv is therefore essential to achieve high yield and reliability [18].

With its rapidly advancing development and even smaller and more complex structures, MST is an interdisciplinary field [19,20,21,22]. It combines micro-mechanics, micro-optics, micro-electronics, biology, and chemistry [20,23,24,25,26]. Manufacturing products, which are usually not intended to be maintained or repaired, involves integrating various manufacturing technologies [27]. These include micro-molding, laser micro-machining, micro-injection molding, surface or bulk micro-machining, lithography, electroplating, additive manufacturing technologies, and molded interconnect devices (MIDs) [20,23,24,25,26,28,29,30,31]. The different manufacturing technologies require a careful selection of the manufacturing process, along with a well-founded consideration and strict control of manufacturing aspects [11,32,33,34,35].

In contrast to macro-production, the consideration of environmental influences and interfaces between the individual manufacturing processes is an essential additional factor in micro-production [10,20,22,29,36,37,38,39]. Due to these circumstances, the relationship between product and process development in micro-manufacturing is significantly more important and often runs in parallel as simultaneous engineering [27,38,39,40,41,42,43,44]. Process development in MST is often a sub-process of integrated product development and must be integrated according to the principle of Design For Manufacturability (DFM) [8,23,24,45,46,47]. The combined development requires both expert knowledge and method-supported knowledge management [48,49,50]. Additionally, individual process steps are often not executed within a single company but are distributed among several small and medium-sized enterprises. This presents an additional challenge and leads to interfaces [51,52,53].

The high complexity of the often numerous process steps requires systematic process control. Even a supposedly small rejection rate in individual process steps can significantly reduce the overall yield [54,55]. To avoid failures and effectively manage the complexity of development, a methodical approach is crucial [9,39,56]. The IEEE International Roadmap for Devices and Systems (IRDS) highlights that further miniaturization and increasing functional complexity are only achievable if substantial improvements in manufacturing yield and economic efficiency are realized. To minimize economic losses caused by yield reductions, the IRDS 2023—Yield Enhancement report emphasizes the growing importance of systematically addressing the process environment, as the list of environmental contaminants requiring control continues to expand [18]. It recommends proactive contamination management throughout the supply chain and the deployment of predictive models at all manufacturing levels to identify critical process parameters that contribute to process fluctuations [18]. These aspects are intrinsically linked to a tightly controlled and stable process environment (PEnv). A systematic integration of these considerations during early-stage process development is essential for reducing yield losses and advancing the economic viability of microsystem technology components. High development pressure causes that this structured development is not always a common practice [36,57]. Due to the wide variety of processes and requirements, there is no standardized methodical approach suitable for all MST process developments [56,58]. In this context, choosing the correct methodology is crucial, as inaccuracies or inadequate execution can lead to development errors [9,36,46,59].

Recent studies show that knowledge management is essential for MST development, but Civelek et al. [34] emphasize its relevance in terms of technology selection rather than consideration of external process influences. In addition to the previously described and well-documented challenges in the literature, environmental influences can have a significant impact on process stability and need to be carefully considered [37,39,60,61,62]. Standards such as EN ISO 9001 require consideration of the process environment (PEnv) to manufacture products of consistent quality in the long term [63].

Following the recommendations of IRDS 2023 [18], this study aims to systematically investigate which process development methods are designed explicitly for MST and to what extent they consider factors of the process environment, as well as which errors can occur despite existing standards and guidelines [64] due to PEnv influences. As this is the first study to address this topic, it provides a broad overview of the existing methods and their characteristics. For this purpose, a mixed methods approach combining a systematic literature review (SLR) and an adapted GRADE (Grading of Recommendations Assessment, Development, and Evaluation) analysis was chosen. In this study, the term “process environment” refers to external influences such as temperature [8,20], humidity [8,38,39], particles [20,23,39], UV radiation [8], vibrations [20,38], and interfaces between processes and humans [38,39,65], which can cause failure and damage [8,20,23,38,39,61,64,66,67]. Especially at the research and development stage, the PEnv must be defined and quantified as an individual product requirement [68]. The methodological consideration of external environmental factors is particularly relevant for research, as it plays a crucial role in ensuring the stability and quality of microfabrication processes. A structured integration of the process environment into process development enables early identification of potential failure sources, increases yield, and sustainably improves the reliability of microcomponents [9,22]. To facilitate comprehension and guide the reader through the structure of this article, the following overview outlines the sequence of sections: Section 1 presents the motivation, current relevance, and objectives of this study. Section 2 provides an overview of external influencing PEnv factors and their potential effects on micro-processes. Section 3 describes the applied mixed methods approach, combining a systematic literature review (SLR) with an adapted GRADE analysis. Section 4 presents and categorizes the methods identified through the SLR, forming the basis for the subsequent evaluation. In Section 5, particularly comprehensive methods of process development are presented and explained in more detail. This should help the reader to better understand the assessments made as part of the adapted GRADE analysis. In Section 6, the adapted GRADE approach is applied to assess the extent to which the identified methods consider the PEnv. Section 7 provides an illustrative example of a micro-process chain affected by environmental influences. Finally, Section 8 summarizes the key findings, discusses practical implications, and limitations of this study, and provides an outlook for future research.

## 2. Overview of Failure Sources Caused by the Process Environment

Different manufacturing processes may have varying requirements, but also impact the PEnv and neighboring processes. This is why micro- and nanotechnology place the highest demands on the control and stability of PEnv conditions. This cornerstone is essential to ensure reproducible results and quality of processes. For this purpose, comprehensive protective measures have been established that precisely regulate temperature, humidity, vibrations, air purity, and electromagnetic interference. Since approximately 1976, standards such as Federal Standard 209B and the current ISO 14644 for cleanrooms have existed, serving as the foundation for such protective measures [64,68]. The DIN EN ISO 14644-4 standard contains a checklist for planning cleanrooms, which includes the requirements for the PEnv and lists a large number of possible influencing factors that can affect processes [64]. Nevertheless, numerous studies show that despite these control mechanisms, quality deviations still occur during production due to environmental influences. Temperature sensors used for cleanroom monitoring, for example, can lose their measurement accuracy due to these influences, which can result in unnoticed changes to the required production environment [69]. A list of relevant sources of failures and their causes is shown in Table 1.

In addition to the mentioned sources of failure, the duration and timing of their impact on the respective process or product are crucial for the occurrence of defects [79]. Due to varying requirements and the effects of the processes themselves, they can also influence the environmental conditions and impair nearby processes [68]. The failure sources and effects listed in Table 1 must be carefully considered when developing new process chains to minimize their impact. This requires early methodological planning and coordination of the individual process steps and their environmental influences.

## 3. Methodology

To analyze existing methods of process development in microsystems technology (MST) and how they consider external influencing factors of the PEnv, the mixed methods approach of Explanatory Design is used [86]. This methodological framework combines a quantitative systematic literature review (SLR) with a qualitative modified GRADE (Grading of Recommendations Assessment, Development, and Evaluation) analysis. The SLR is used to identify relevant methods for process development in MST. After identifying the methods, a systematic evaluation is carried out regarding the consideration of the PEnv using a modified GRADE analysis. The procedure for conducting the SLR and GRADE is described below.

### 3.1. Systematic Literature Review (SLR)

The systematic literature review (SLR) is considered one of the most efficient techniques for conducting a comprehensive literature analysis [87]. In contrast to a traditional literature review, an SLR provides a structured approach that allows for the literature to be searched and evaluated according to predefined relevance criteria [88]. The systematic search and comparison of the retrieved literature increase the reliability and quality of the results compared to conventional literature reviews [87]. The SLR is divided into three main phases: the initial phase involves planning the systematic search, followed by the execution of the search, and finally the documentation and evaluation of the research results [89,90]. The procedure in these three phases is explained in detail in the following subsections.

#### 3.1.1. Phase I: Planning Systematic Search

The planning was carried out using the PRESS and the PRISMA 2020 checklist, which supports the preparation of a systematic literature search [91,92,93,94,95,96]. This review was not registered in a systematic review database. However, a research question, a structured outline, and a search design were developed in advance and are reproduced in this paper. At the beginning, the precise research question is defined to outline the objective of the systematic literature search clearly. The research question in this study is: “What methods are available for process development in MST, and can they be used to consider the external influences of the PEnv?” Based on the research question, relevant subject areas and corresponding keywords were derived to identify the primary literature. The list of keywords is provided in Table 2. Subsequently, inclusion and exclusion criteria were established to filter out irrelevant studies. The criteria are shown in Table 3.

#### 3.1.2. Phase II: Carrying out the Systematic Search

The procedure for systematic literature searches is based on the classical structure and was adapted to include the use of Artificial intelligence (AI) search tools [88,96,97,98]. The search is delineated into eight structured steps, the sequence of which is illustrated in Figure 1. Several search chains were initially constructed from the specified keywords using Boolean operators. These search chains were the foundation for exploring various literature databases in the subsequent step. The methodological approach for literature research included both conventional database searches and AI-supported searches. This two-part approach enhances the reliability of evidence by contributing to the comprehensiveness and robustness of the literature search, thereby establishing a broader foundation of explored literature. First, traditional literature databases were searched using the established search chains. Subsequently, AI tools were employed for further literature searches, whereby the pre-established search chains were converted into prompts optimized for AI utilization. The specific databases and AI tools utilized are enumerated in Table 4.

In the third step, a primary selection is conducted based on the titles. From the totality of the primarily selected literature, duplicates are eliminated in the fourth step. The fifth step involves excluding literature based on abstracts and predefined exclusion criteria. Using the AI tool Litmaps, the sixth step examines the references of the remaining literature for additional relevant publications. Newly identified sources are subjected to selection again, starting from the fourth step. The seventh and final step involves selecting the relevant literature based on the full text. After completing this systematic selection process, a final selection of the relevant literature was included in this SLR.

The risk of bias in this SLR exists due to the use of citation networks and AI-based tools. Citation networks tend to overrepresent frequently cited studies, which can lead to relevant but less well-known studies being overlooked. In addition, AI-based tools such as Perplexity or ChatGPT 4o often provide slightly different results for identical search queries, making the reproducibility of the search process more challenging. There is also a risk of algorithmic influence on search results, favoring certain sources and potentially biasing the selection of relevant studies. However, due to the additional and extensive literature search in classical literature databases, the risk of bias due to missing results can be assessed as low. Another potential risk of bias is that this study only analyses methods explicitly developed for the requirements of MST. Generally applicable methods that can also be used in MST are not considered. These include, for example, material flow planning methods, whose logistical planning character does not necessarily require specific adaptation to MST.

#### 3.1.3. Phase III: Documentation and Evaluation of Relevant Methods

The selection process for the reviewed literature is illustrated in a PRISMA flow diagram, as shown in Figure 2 [92]. The process begins with an initial selection of articles broadly relevant to the research topic. Literature with abstracts aligned to the topic and not excluded by predefined criteria was examined through a citation network to identify additional relevant studies. The amount of scientific literature was continually reduced based on selection criteria, resulting in a final selection of the relevant literature for this study. A list of the relevant literature, organized by application area, is provided in Section 4.

### 3.2. Modified GRADE Analysis

The GRADE system (Grading of Recommendations Assessment, Development, and Evaluation) was initially developed for the medical field to assess the quality of evidence and the strength of recommendations in systematic reviews and guidelines [99,100,101]. The study by Bilotta et al. demonstrates that GRADE, due to its structured and transparent evaluation methodology, is also suitable for assessing evidence regarding environmental influences [102]. The authors specifically modified the GRADE system to align with the engineering methods and terminology used in this study, examining its suitability for considering the PEnv in MST process development. The assessment of the evidence made here relates exclusively to the suitability for consideration of the PEnv and says nothing about the quality of the evidence for the primary purpose of the methods. The analysis includes five modified criteria: risk of bias, indirectness of evidence, imprecision of results, adaptability to the PEnv, and publication bias. The evidence of the results is then categorized into four quality levels. A detailed explanation of the modified criteria and quality levels is provided in Section 6. The evaluation results enable a systematic classification of existing methods and identify potential research gaps regarding the consideration of PEnv in MST. First, however, in Section 4 and Section 5, the methods identified in the SLR and their categorization are presented. 

## 4. Overview of Process Development Methods

This section presents which methods available specifically for process development in MST, their purpose, and whether these methods are also suitable for incorporating PEnv into process development.

Table 5 provides an overview of methods for process planning in MST identified as part of the SLR. The methods listed are divided into two main categories: software-based methods and framework-based methods. The table refers to methods that cover all or only some aspects of process development in MST. If methods cover several aspects of process development, these are listed in each category for which they are applicable.

Comprehensive methods that cover the entire process development cycle are described in detail in Section 5 to better understand the subsequent evaluations in the GRADE analysis. None of the methods found covers the comprehensive consideration of the PEnv. Therefore, methods that allow for considering the PEnv, if actively adapted by the user, are included. This also includes generally applicable methods such as Process Failure Mode and Effect Analysis (PFMEA), in which the user can actively incorporate aspects of the PEnv, provided they are known.

## 5. Methods for Process Development in Microtechnology

The following section presents comprehensive methods covering large parts of the process development cycle in MST. It provides an overview of their characteristics and supports the understanding of their evaluation in the GRADE analysis in Section 6. The methods described here were selected based on their multiple publications and citations in the scientific literature or current research, which underlines their relevance in this field. This section aims to give the reader in-depth insight into the various approaches and tools used to optimize and control the production process of microsystems. In particular, the integration of product and process development, the use of CAD systems, the management of production data, and the use of simulation and modelling techniques are discussed. The methods presented provide a comprehensive framework for the efficient and precise development of microsystems, considering both framework- and software-based solutions. The comparison Table 6 summarizes the characteristics of the methods with a quick overview.

### 5.1. Microspecific Product Development Process for Tool-Based Microtechnologies (μPEP) (2005)

The microspecific product development process (μPEP) for tool-based microtechnologies is a method for developing microsystems that also integrates the parallel planning of the manufacturing process [40]. This approach is based on systematic, iterative development from the conception of requirements to prototyping and combines basic design rules with the specific requirements of microtechnological manufacturing processes [104]. The basic procedure of the μPEP method is described in the sickle model in Figure 3, which is conceptualized based on the Y-model developed further by Walker and Thomas in 1985 [147]. The sickle model is divided into three concentric phases: The sickle model is divided into three concentric levels: the structure level, the component level and the system level. These three levels are completed by the validation and prototyping phase [104].

This model enables the application of both top-down [148] and bottom-up development approaches, whereby system-wide requirements can be transferred into detailed technical specifications and innovative solutions can be developed from the available materials and technologies [107,149]. The microspecific product development process (μPEP) thus offers a fundamental approach to developing microsystems. The manufacturing process is developed with the aid of CAD systems with linked knowledge management [40,105] and databases for production data [103,106]. The application of this method requires a deep understanding of the different manufacturing processes, technological influences, and limitations [149]. This model does not consider interfaces between the different manufacturing processes, material flows, or external environmental influences that can affect the manufacturing process. Therefore, additional analysis and optimization tools are necessary to implement a comprehensive development and manufacturing strategy that considers all relevant factors.

### 5.2. Adapted V-Model for MEMS Development (2005)

The method is designed for holistic product development in MST and is based on an adapted version of the V-Model of the VDI 2206 standard [10,27,41,57]. Due to the high importance of manufacturing aspects in MST, the method is focused on the parallel development of the product and manufacturing process in interdisciplinary projects [27,150]. This includes the selection of suitable manufacturing technologies at the beginning of the development as well as the continuous adaptation of these, considering design rules [10,36,56,57]. The method’s structure consists of four basic elements. They are divided into the procedure model, the development procedure, detailing, and the use of supporting methods. The V-shaped process model, shown in the center of Figure 4, supports the design in the concept development phase [10,27,41,57]. In the development process, the V-Model runs through several iterative development loops [10,27,56]. The following detailing covers the entire development procedure of the V-Model shown in Figure 4 and includes different perspectives on the development procedure [27,56,150]. Watty et al. provide a checklist to take the complex requirements into account [57]. This checklist includes selecting a suitable PEnv, reviewing set standards or transportation conditions. To overcome development-related challenges, the appropriate use of supportive methods is recommended, which may be either specific to MST or interdisciplinary [10,56].

Overall, the methodology offers a structured and systematic approach characterized by an integration of various business disciplines and supporting methods. Despite its flexibility, the method does not provide a precise approach for selecting a suitable manufacturing process and defining the necessary PEnv. This planning aspect must be compensated by supporting methods or expert knowledge. The need for resources (time, personnel, and coordination) to implement this comprehensive methodology is thus a challenge.

### 5.3. Software-Supported Process Development Execution System (PDES) XperiDesk (2008)

The Process Development Execution System (PDES), supported by the commercial software program XperiDesk, forms a technology management system for process development [13,115,117]. The PDES is based on the pretzel model, which was developed for the integrated development of product and manufacturing processes in MST [43,51,114,116]. The model shown in Figure 5 is characterized by the intertwined design procedures of a behavior-driven top-down synthesis process and a technology-driven bottom-up analysis process [14,42,53,116]. This structure enables an iterative development that effectively describes and structures the development process of MEMS.

The knowledge base of production-relevant data generated during the development cycles enables further development cycles to be accelerated, as long as the knowledge is systematized and managed [24]. XperiDesk software is used to manage this knowledge base and the practical application of the PDES by providing essential management and execution functions [13,117]. The scope of the software illustrated in Figure 6 facilitates the development process through structured management of technology data, material information, design rules, process steps, and their interfaces [13,14].

Its strength in the context of PEnv lies in the combination of a knowledge database [53] and an integrated simulation interface [13,14]. All process-relevant parameters can be stored as data in the PDES and included in a simulation interface. This can also be specifically expanded by the user to include PEnv data. In practical terms, this means that developers can define cleanroom class, temperature, or humidity as process parameters, for example, and check their influence. In early development phases, a PDES allows for virtual verification and optimization of processes under different environmental conditions even before real tests are carried out. This forward-looking simulation of “what-if” scenarios is particularly advantageous in order to be able to react quickly to changes in the environment (such as stricter cleanliness requirements or a different plant environment). In the prototype phase, a PDES also offers a tracking environment that monitors real experiments and documents all environmental conditions [13]. This makes deviations between virtual planning and real execution transparent, and process development can be continuously improved. Compared to the other methods, a PDES in conjunction with XperiDesk is characterized by a deep integration of PEnv data via digital knowledge management and simulation, which enables a more accurate estimation of environmental influences. However, the prerequisite for this is that the relevant influencing factors are known and consciously recorded in the database by the user.

The PDES, in conjunction with XperiDesk, provides small and medium-sized enterprises (SMEs) with a platform to efficiently develop manufacturing processes, facilitate data exchange, and adapt these processes to specific product requirements [24,43,51,52,53,116]. The use of technology data from previous developments, as well as an integrated consistency check, can accelerate future processes and reduce the dependence on expert knowledge [13,24,53]. Despite its advantages, the software requires an extensive, continuously maintained database, which leads to costs and maintenance efforts. Without regular updates, incorrect decisions can impair effectiveness. Although the software has been continuously developed since 2008 [117] and is still available (as of May 2025) [152], the lack of current publications indicates a lack of market penetration.

### 5.4. Micro-Process Planning and Analysis (µ-ProPlAn) Method (2013)

The “Micro-Process Planning and Analysis” (µ-ProPlAn) method is a process planning method that is used to plan and design process chains in MST. Its primary focus is modelling and analyzing the interdependencies between the various production technologies and the logistical interfaces between the individual production steps [33,58,126,129,130,131]. The method is characterized by a holistic view of the process and material flow planning, the configuration and design of the processes, and the evaluation of the modelled process chains using process simulations and statistical methods [130]. Following the simultaneous engineering approach, the methodology enables a detailed analysis of process efficiency to be carried out as early as the product planning phase, weak points to be identified, and these to be optimized using suitable process parameters [33,58,126,130]. The method can also be used to evaluate existing process chains [58].

The method addresses key challenges in micro-process development, such as low tolerances, size effects, and limited process or machine availability [125,126,128,129,130,131]. As illustrated in Figure 7, the µ-ProPlAn method is divided into two main phases: modeling and evaluation of the production system. Modeling includes process chains [126,129,130,153], material flows [126,129,144], and cause–effect networks [126,127,129,130] based on relevant data and mathematical methods [58,126,129,130,131]. Generating detailed cause-and-effect networks also enables the consideration of process interfaces and external influences of the PEnv. However, this requires specialized expertise and a profound understanding of key parameters. The application of the method allows for a high level of flexibility, for example, by inverting cause–effect networks to plan a defined output [125]. In addition, a process chain design can be carried out using geometric product parameters [58,127,128]. Once modeling has been completed, the various process chains are evaluated and assessed based on technical feasibility and logistical performance [58,129,130]. These must be continuously reviewed based on expert knowledge and new data [112]. A further function of μ-ProPlAn is change propagation. Based on known or calculated process data, this function allows for the estimation of the effects of parameter changes along the entire process chain. In theory, this can also be applied to variable influences of the PEnv. As a result, it is possible to simulate during the planning phase how a sudden variation of an environmental parameter affects subsequent process steps and the final product, enabling the proactive development of countermeasures [128]. Change propagation can also be applied to existing processes to simulate fluctuating process parameters and thus find errors in processes. So far (as of May 2025), only one prototype exists for a software-supported method application [58].

In summary, µ-ProPlAn supports efficient manufacturing process design by integrating multiple modeling perspectives. It provides detailed planning results regarding process flows, logistics, manufacturing quality, production time, and proactive consideration of possible disruptive influences without needing a physical prototype. However, the method is resource-intensive and demands expert knowledge and mathematical proficiency [112,126]. Although the method is not primarily intended for this purpose, it can also consider influencing factors of the PEnv, which also requires expert knowledge. When applied correctly, µ-ProPlAn offers a strong foundation for evaluating and iteratively optimizing process chains.

### 5.5. Product Development Methodology for iMST (2023)

The method describes an approach to developing microsystems that mainly focuses on the specific requirements and challenges in producing individualized microsystem technologies (iMST). As shown in Figure 8, the method offers a comprehensive strategy integrating design rules and manufacturing considerations to enable efficient and flexible production. However, process interfaces and external environmental influences on the process are not considered. It emphasizes the importance of closely interlinking product development with the possibilities and limitations of manufacturing technology to optimize the performance and manufacturability of the end products.

Depending on the specific conditions of each development stage, the use of supporting methods is recommended. The method’s structure is divided into different phases, ranging from conceptualization through design to prototype development and subsequent production. Each phase uses specific tools and techniques to increase development efficiency and ensure product quality. The method uses advanced modeling tools and simulation techniques to make design decisions early in the development process. The modern integration of Co-design CAD (MCAD and ECAD) systems and the application of simulation tools allow for quick reactions to design changes and their efficient implementation [34,48].

When planning the manufacturing process, particular emphasis is placed on considering all relevant factors, such as material selection, manufacturing processes, and assembly techniques. The KETTS-V0.2 (Knowledge-Engineering Tool for Technology Selection) software program was presented as a supporting method for selecting suitable manufacturing processes. The framework, whose source code is openly accessible, supports experts in knowledge management but simultaneously reduces the dependency on expert knowledge. The software can systematically support selecting suitable manufacturing technologies and help compare process capability and product requirements. This is especially important for effectively facilitating the individualization of microsystems and ensuring high product quality. Material flows and interfaces to the manufacturing technologies, such as environmental parameters, are not considered with this method, which is why other supporting methods should be used for this purpose [34,48].

### 5.6. Comparative Overview of Selected Process Development Methods in MST

To improve clarity and facilitate a structured comparison of the analyzed process development methods in microtechnology, Table 6 below provides a concise overview. It contrasts key methodological characteristics and helps the reader quickly understand the differences and relationships between the various approaches.

The column “Application in the development process” indicates the application of process development in which the respective method is mainly to be used. “PEnv explicit?” denotes whether the method explicitly considers the process environment or leaves this to the discretion of the user. “Data-based?” captures whether the method relies on or systematically integrates concrete process data. “Model-based?” indicates whether the methodological approach is built on structured models such as cause–effect networks or simulations. The final column, “Main advantage”, summarizes the specific strength of each method concerning process development, particularly in view of environmental conditions. This comparative overview serves as an orientation for method classification and forms the basis for the subsequent qualitative assessment.

In summary, the methods analyzed follow different approaches to process development, ranging from rule-based (μPEP) to model-based (μ-ProPlAn) and database-supported systems (PDES). The consideration of the process environment varies between explicit integration (e.g., via checklists in the extended V-model) and implicit handling through modeling or simulation. Depending on the application scenario, each method offers specific strengths, which will be systematically evaluated in the following sections.

## 6. Analysis of the Categorized Methods Using the Adapted GRADE System

According to the GRADE system, the analysis was conducted using the method categories listed in Table 5, which were identified as part of the SLR. The quality of evidence was analyzed based on five evaluation criteria, and each of these was then evaluated according to four different quality levels of evidence. The evaluation criteria were adapted to fit the topic.

The evaluation criterion “Inconsistency of results” was replaced, as it refers to the variability and contradiction of results between different studies. Almost none of the examined methods explicitly consider the PEnv, so no results are available for comparison. Instead, a new evaluation criterion, “Adaptability to the process environment”, was introduced. It is used to assess how flexibly the methods can be adapted to the specific requirements of the PEnv. In Table 7, the overall GRADE evaluation criteria of this study are explained. This analysis enables the identification of the strengths and weaknesses of the various categories of methods, while also highlighting existing gaps in the literature.

### 6.1. Quality Levels of Evidence: Structure and Application

The GRADE system classifies the evaluation criteria based on the quality of their evidence into four quality levels, facilitating a transparent and standardized assessment of confidence in the results. These quality levels are the same for all five evaluation criteria in this study and are presented in Table 8. However, their qualitative interpretation varies based on the specific context of each criterion.

### 6.2. Evaluation of the Method Categories According to Modified GRADE Criteria

In the following section, the methods of the individual categories are analyzed, compared, and evaluated according to the modified GRADE system. Following the detailed analysis of the five criteria, each method category is subjected to an overall qualitative assessment. A differentiated classification is used to determine the extent to which the results of individual evaluation criteria impact the evidence of the entire method category. The overall results of the analysis are presented as a “Summary of Findings” in Table 9.

#### 6.2.1. Risk of Bias

Production-oriented product development

Methods in this category primarily focus on optimizing product designs and fundamental considerations of manufacturing processes. However, an explicit consideration of PEnv conditions is not included [34,104,109]. An exception is the method of Watty and Binz [56,57], which at least indicates the relevance of process environmental factors with a checklist. This checklist makes it possible to systematically define critical environmental conditions at the start of the project. This ensures that the necessary cleanroom classes, temperature and humidity limits or transportation conditions, for example, are documented as requirements during the concept phase. Explicitly recording the PEnv factors at an early stage is an important advantage because it prevents environmental aspects from being taken for granted. In contrast to other methods, this approach reduces the risk of overlooking PEnv influences and lays the foundation for taking necessary infrastructural measures or standard specifications (e.g., ISO 14644 for cleanrooms) into account from the beginning. How detailed environmental factors are actually included depends heavily on the user and any additional methods used (e.g., a Process FMEA with a focus on PEnv).

Implicitly, process environments can be integrated into methods that deal with defining and evaluating process parameters. Examples include the PDES method [24] and the approach by Gleadall et al. [15]. In these cases, specific requirements for the PEnv can be defined as process parameters, but this is not explicitly intended and requires active consideration by the users. The approach of Sagoo et al. [109] includes a testing and verification cycle that could theoretically also consider the influences of the PEnv. However, this is only carried out after the product has been manufactured. The implicit or missing consideration carries the risk that the PEnv is assumed to be ideal without considering the influences that are present in reality. Unrecognized sources of error in early development stages can impair the efficiency and quality of manufacturing processes. Considering the increased risk of methodological bias, the evidence of the results in this category is rated as “low”.

Development/planning of process chains

The systematic integration of essential environmental parameters is crucial throughout the entire development cycle, whereby their impact is most significant in process chain development. Interfaces between the process steps are only considered in the method of Vella et al. [22], the PDES [13,14], and µ-ProPlAn [126,127,129,130], focusing exclusively on technological interfaces between the processes. The PEnv is not explicitly considered. As a result, undetected disruptive factors can impair the overall process and product quality. Approaches such as the modified V-Model by Watty and Binz [56,57], the PDES [24,52], or the µ-ProPlAn method [126,129] only offer the possibility of integrating relevant PEnv parameters after targeted expansion by the user. The PDES and µ-ProPlAn methods offer the possibility to simulate process chains under varying environmental conditions based on existing PEnv, thereby reducing the risk of unexpected yield losses. However, as this approach requires a targeted extension of the methods, as previously discussed, it cannot be considered common practice. Other methods in this category only consider the process flow in an idealized way and do not systematically integrate PEnv parameters [9,34,50,120,121,122,132]. This increases the method category’s bias risk, so the evidence is rated as “low”.

Selection of manufacturing processes

Methods such as those of Zha et al. [133] or Rippel et al. [127] compare manufacturing processes based on various process-specific requirement values, thus enabling implicit consideration of the PEnv. Leaner methods, such as the KETTS software program from Civelek [48], focus exclusively on manufacturing properties and do not consider the PEnv. Depending on the product, this reduced focus may be sufficient but carries a moderate risk of bias if PEnv influences are ignored. This is particularly relevant for manufacturing processes that generate PEnv influences affecting subsequent process steps or product quality. For example, a particle-generating laser-cutting process can lead to inclusions in later coating processes or impair the functionality of micro-optics. Due to these limitations, the evidence quality for this category is rated as “moderate”.

Material selection

Methods for material selection focus on technical and product-specific requirements, while PEnv influences such as humidity or UV radiation are often not considered [135,137,140,141]. This leads to a moderate bias risk, which arises less from the material selection itself and more from its practical handling. Therefore, the evidence is rated as “moderate”.

Material flow planning

In material flow planning, a distinction must be made between material flow optimization and considering transport and storage processes. The first focuses on processing times and logistics chains and carries no risk of bias, as the PEnv does not influence this. In contrast, transport and storage processes present significant bias risks due to interfaces, human influences, and storage conditions. Thematically, these aspects belong more to process chain planning and are rarely considered in material flow planning [23,44,48]. Only µ-ProPlAn [126] and the PEDS [24,43] take both approaches into account, whereby the PEnv must also be consciously included here. However, the focus of material flow planning methods is clearly on material flow optimization [23,24,43,44,48,126]. There is no systematic risk of bias here, as the PEnv does not influence this. Consequently, this category’s evidence is rated as “moderate”.

#### 6.2.2. Indirectness of Evidence

Production-oriented product development

Product development methods in MST that include the manufacturing process do not systematically incorporate the PEnv. It is crucial to integrate suitable production conditions with regard to manufacturability and product failure minimization. In current approaches, the conditions prevailing in practice are often only checked implicitly using production factors that have to be selected by the user [15,24,47,57] or are not considered [34,104,109]. Therefore, the evidence of the results regarding the PEnv is indirect and rated as “low”.

Development/planning of process chains

Methods in this category primarily focus on the technological and logistical planning of process chains. Process environmental factors are only rudimentarily considered or assumed to be static. Therefore, the methods can only be used indirectly to analyze the PEnv. Methods such as those by Vella et al. [22], µ-ProPlAn [126,129], or PDES [24,52] offer approaches for integrating various influencing factors but strongly depend on user customization. Without explicit customization, none of the methods is able to take the PEnv into account systematically. Due to the only indirect possibility of including the PEnv, the evidence is classified as “low”.

Selection of manufacturing processes

When selecting manufacturing processes, the PEnv is only implicitly [127,133] or not at all considered [23,47,48]. This is because the systematic inclusion of PEnv influences is rarely required for the primary purpose of the methods. This means that the methods are only suitable for this purpose to a limited extent. This results in a high level of indirectness of the results, which means that the evidence is rated as “very low”.

Material selection

The methods investigated are indirect, as they are primarily focused on isolated material properties in standardized process environments. Popular approaches like the Ashby method do not consider PEnv influences [135,137,138,139,140,141,142,143,154]. More flexible methods, such as VIKOR and TOPSIS [140,141], as well as the approach by Zha et al. [133], theoretically allow for integrating PEnv requirements into their decision matrices and requirement values. However, this requires active customization by the user, which is not mentioned in the literature. Consequently, the methods are not directly suitable for considering the PEnv, so the evidence is rated as “very low”.

Material flow planning

The analyzed method category shows a high level of indirectness in considering the PEnv in material flow planning. The focus is primarily on temporal and quantitative availability, while PEnv factors remain unconsidered. Therefore, the evidence is rated as “very low”.

#### 6.2.3. Imprecision of Results

Production-oriented product development

The integration of the PEnv into existing methods is either only implicit or wholly neglected. Therefore, the results show low precision regarding product-related requirements for the PEnv. In methods such as PDES [24] or the approach by Gleadall et al. [15], users with a high level of expertise can define and quantify PEnv requirements as necessary process parameters, but this requires a methodological extension. The level of detail achieved varies depending on the user’s expert knowledge. Consequently, the results of this method category remain imprecise regarding the PEnv. Therefore, the evidence is classified as “very low” according to the evaluation criteria.

Development/Planning of process chains

Integrating product-specific PEnv requirements is particularly crucial for process chain development’s process security. Precise data, such as particle limit values, humidity, electrostatic charge, or external influences at interfaces, are essential. Methods such as µ-ProPlAn [126,129] and the PDES [24,52] can provide more precise results through cause-effect networks and defined process parameters. However, this requires high-quality input data and knowledge of potential critical process influences. Without these data, accuracy remains limited. The application of the method by Watty and Binz [56,57] only provides basic indications for considering the PEnv. The level of detail of the results depends on the user and additional supporting methods. Methods that ignore external influences or assume the PEnv to be ideal produce the most inaccurate results [9,22,34,50,66,120,121,122,132]. Accordingly, the evidence of the results is rated as “low”.

Selection of manufacturing processes

Due to their significantly different focus, these methods barely consider the PEnv. As a result, the outcomes of investigations regarding the PEnv are imprecise. Methods in this category are not suitable for this purpose. Therefore, the quality of the evidence achievable with these methods is rated as “very low”.

Material selection

The investigated methods provide precise and structured results for specific material properties determined under standardized conditions. The PEnv can be implicitly considered via material parameters such as manufacturer specifications for humidity and temperature during storage and processing. This information enables a quantifiable adjustment of the PEnv. However, these methods do not systematically consider the PEnv in their regular application. Accordingly, the evidence regarding PEnv influences is reduced to “moderate”. A targeted expansion of flexible methods to critical PEnv influences can significantly improve the evidence.

Material flow planning

The PEnv plays no role when focusing on operational parameters such as lead times and material availability. This focus on methods for material flow planning [23,24,43,44,48,126] means that the evidence of the results in relation to the PEnv is rated as “very low”.

#### 6.2.4. Adaptability for Considering the Process Environment (New Category)

Production-oriented product development

As mentioned in Section 6.2.1 and Section 6.2.2, the requirements of the PEnv can be incorporated into some methods if they are defined and quantified as process parameters [15,24,47]. However, this requires the user to identify critical factors through expert knowledge or knowledge management. These factors vary depending on the product and application. It is not possible to adapt the other methods in this category without significantly expanding the methodological scope. Therefore, the evidence of achievable results after adaptation for considering the PEnv is rated as “low”.

Development/Planning of process chains

In µ-ProPlAn [126,128,129], the simulation of varying PEnv influences on process quality could be carried out using cause–effect networks and change propagation. In PDES [24,52], necessary PEnv factors and process data can be included in the XperiDesk knowledge database and evaluated simulatively if required. This results in a practical advantage: Known environmental requirements (such as humidity or cleanliness specifications) are incorporated directly into the process development procedure and can be checked for their effects using simulation. In this way, the PDES can react efficiently to changing environmental conditions, particularly in iterative development cycles when adapting existing processes, and provide decision-making aids without having to rely exclusively on expert knowledge.

Watty and Binz [56,57] recommend, depending on the application, the targeted extension of their extended V-Model with supporting methods. For this, a PFMEA focusing on the PEnv [145,146] would be suitable. Such independent methods can also complement process planning methods that do not systematically consider the PEnv. Two further possibilities are offered by the method of Vella et al. [22], which uses a Process Pair Maturity Matrix (PPMM) and an expert questionnaire to assess technological maturity and reproducibility. Aspects of the PEnv could be integrated into the PPMM or considered by involving relevant experts in the survey. However, for effective application, it is necessary that the user consciously expands the methods to include relevant influencing parameters. Otherwise, consideration of the PEnv remains limited. The evidence of the results is rated as “moderate,” provided the user has sufficient expert knowledge to identify critical PEnv influences.

Selection of manufacturing processes

Some of the investigated methods offer the theoretical possibility to integrate PEnv parameters due to their knowledge-based approach [127,133]. The extent to which this is implemented depends on the user of the method. Most methods in this category are not designed to consider PEnv factors. Due to their limited adaptability, the evidence for this category is rated as “low”.

Material selection

Methods such as VIKOR, TOPSIS [140,141], or the approach by Zha et al. [46,133] are characterized by flexibility. A targeted expansion to include critical PEnv factors is not intended, but would be possible. Methods purely focused on material parameters, such as the Ashby method [135,137,138,139,140,141,142,143,154] or the approach by Shukor et al. [9], are difficult to expand. Therefore, the evidence that can be obtained by adapting the methods is rated as “low”.

Material flow planning

If the PEnv is considered in material flow planning methods, it only makes sense for those methods that incorporate physical material transport along with logistical optimization. These include methods such as µ-ProPlAn [126] and PDES [24,43], which can be adapted to consider, for example, process environmental influences during storage or material transfer between different environmental conditions. Another approach is indirectly described by Niekiel et al. [44], where a part of each batch is partially processed in a test run. If quality differences occur, a process adjustment is made. This approach can also be applied when material parameters are affected by the PEnv. Due to these possible adaptations, the evidence for methods incorporating physical material transport is rated as “moderate”.

#### 6.2.5. Publication Bias

Production-oriented product development

The examination of this category shows that methods for product development in MST rarely systematically incorporate product-specific PEnv requirements. As a result, disruptive influences can be overlooked. This gap in consideration may be due to the fact that the methods for product development are already extensive. The additional inclusion of a technically complex topic, such as PEnv, can make it more challenging to apply in practice. Systematic fundamental considerations regarding product-specific environmental requirements for production are nevertheless important to consider quality-changing influences at an early stage. This bias in existing methods significantly reduces the quality of evidence in relation to the PEnv. Therefore, it is rated as “low”.

Development/Planning of process chains

The majority of the analyzed literature focuses on the technological planning and optimization of process chains as well as logistical aspects. A publication bias exists for methods that systematically incorporate the PEnv. In practice, necessary PEnv factors are only implicitly considered through expertise or production standards. However, this implicit approach carries the risk of missing potential sources of error, which reduces the evidence of the results and limits the transferability of the findings to real process conditions. In this process development stage, neglecting product-specific PEnv requirements carries the most significant risk for quality fluctuations. Due to the publication bias, the evidence for this category is rated as “very low”.

Selection of manufacturing processes

The lack of consideration of the PEnv in these methods results from its minor influence on the manufacturing process selection. With a few exceptions, such as integrating new processes into an existing production environment, the consideration of the PEnv does not bring any benefit here. Due to the low publication bias, the evidence is rated as “moderate”.

Material selection

Approaches such as the Ashby method [135,137,138,139,140,141,142,143,154] are widely used because they are intuitive, well documented, and easy to apply. However, their focus is almost exclusively on material properties under idealized conditions, without considering the PEnv. More flexible approaches, such as VIKOR and TOPSIS [136,140,141], are also frequently found in the literature but are mainly applied to classical multi-criteria decision problems. There is no explicit integration of PEnv parameters. This indicates a slight preference for established more straightforward approaches. Therefore, the evidence quality is downgraded to “moderate”.

Material flow planning

Due to the focus on logistical efficiency, which is independent of the PEnv, there is a very low risk of publication bias. Therefore, the evidence is rated as “high”.

### 6.3. Overall Evaluation of the Method Categories and Summary of Findings

Production-oriented product development

The evaluation shows that the method category has weaknesses in systematically considering the PEnv. The lack of systematic consideration of product-specific PEnv requirements presents a bias risk that can lead to quality deviations in practice. Therefore, the overall quality of evidence is rated as “low”.

Development/Planning of process chains

Overall, it can be observed that the extensive methods for planning process chains have significant weaknesses when considering the PEnv. The analysis and definition of process-specific PEnv requirements are particularly relevant in this development phase. On the one hand, the possibility of adapting the methods is most feasible here. However, at the same time, the publication bias is exceptionally high due to the previously missing consideration. The overall quality of evidence is rated as “low”.

Selection of manufacturing processes

The methods for selecting manufacturing processes are designed to provide robust results for their specific application. The evaluation shows that these methods are barely or not suitable at all for analyzing the PEnv. The overall quality of evidence is rated as “very low”.

Material selection

The method category provides solid approaches for evaluating and selecting suitable materials based on technical properties. However, it does not explicitly consider PEnv factors such as temperature, humidity, cleanliness, or UV radiation. More flexible methods, such as VIKOR and TOPSIS, are theoretically adaptable but require active user adjustments and additional data. Therefore, the overall quality of the results is rated as “low”.

Material flow planning

Methods in material flow planning primarily focus on logistical efficiency. PEnv influences are hardly relevant to the primary purpose of these methods. Therefore, they are not systematically considered, and the adaptability of these methods is limited. Consequently, the evidence quality regarding the PEnv is rated as “very low”.

## 7. From Theory to Practice: The Influence of External Factors on an Example Process

Considering the PEnv in the development of manufacturing processes in MST is essential to identify potential influences that could lead to component defects at an early stage. To illustrate the relevance of these influencing factors, an exemplary manufacturing process for a microstructured injection-molded part is analyzed below. The example process is based on an actual industrial application but has been abstracted and anonymized.

The process includes the production of an injection-molded part with microstructures (<5 µm), which is scanned by a micro-optical component in its final application. High requirements are demanded of the microstructure in terms of precision and freedom from defects. The process chain consists of several spatially separated steps, as shown in Figure 9. First, stored raw material is fed into the machine and dried. Next, the micro-injection-molded part is produced using the injection molding process. The parts are then packaged and transferred to the following process stage of cleaning in a plastic transport container. After cleaning in an ultrasonic bath, the components are repackaged and then undergo a sputtering process for surface coating. The coating is followed by quality control and final packaging.

The necessary transfers among the various process steps expose the components to different environmental influences. Table 10 presents possible PEnv influences, potential component defects, and the thematically related method categories. The example shows that even a relatively short manufacturing process in MST can be influenced by various PEnv factors. Influences from the PEnv do not always have an immediate negative effect, but can also impact subsequent processes. This highlights the necessity of a holistic approach. Therefore, it is insufficient to consider process conditions in an idealized manner. Actual conditions must be systematically integrated into process development to ensure product quality and process stability.

Established methods such as the Process FMEA or the μ-ProPlAn method can be used to systematically consider potentially negative influences of the PEnv as early as the planning phase. In contrast to PFMEA, μ-ProPlAn enables model-based integration of influencing variables into the design process. The cause–effect networks in conjunction with the prediction of variations through change propagation [128] can serve as an interface for displaying parameters of the PEnv (e.g., cleanroom class, ambient humidity, electrostatics) and their effects on process variables. In this way, it is possible to check how sensitively a process reacts to certain environmental changes as early as the development stage. If it is recognized, for example, that a variation in the ambient humidity or the number of particles influences the quality of a workpiece or the process yield, this can be stored in the model and taken into account in the process design. However, the prerequisite for reliable predictions is closely linked to reliable data and existing expert knowledge about critical and relevant process variables. Real-time monitoring can then be used to react to changes by adapting the process or implementing previously defined countermeasures. This interplay of knowledge model (μ-ProPlAn) and specific production data ensures that sudden changes in the environment can be responded to quickly and effectively before they lead to rejections or failures.

## 8. Discussion and Outlook

Despite existing standards and controlled environmental conditions such as clean rooms, PEnv influences in MST processes can still lead to failures and quality deviations. Zhou et al. [61,67,74,75,76], Soueid et al. [68], Wolpert et al. [70], and others describe these failure mechanisms in their studies. Addressing these specific and individual process error sources necessitates a focused adjustment of the PENv. For efficient process development, it is essential to recognize and avoid these failure risks during process development. However, the results of this study show that existing MST process development methods do not adequately consider the requirements of the process environment. The lack of methodological integration reveals a research gap.

### 8.1. Publication Bias and Indirectness of Evidence

The GRADE analysis of all methods explored in the systematic literature review (SLR) shows that they are only indirectly suitable for analyzing the PEnv, as their primary application focuses on other aspects of process planning. An independent method for analyzing and planning a process-specific PEnv in MST does not exist. The limitations of this study lie in the fact that only methods designed explicitly for MST were examined. The only explicit consideration in the methods of MST can be found in the adapted V-Model by Watty and Binz, which also includes a checklist considering the influences of the process environment [57]. Most partial methods for material selection, material flow planning, and process selection, as well as the holistic methods (PDES [13,24,116], µ-ProPlAn [112,127,129], iMST [11,34,48], μPEP [104,107]) do not explicitly consider the PEnv in the planning process. This carries the risk that the process environment is assumed to be ideal or static. The lack of systematic consideration of the PEnv in existing methods reduces their transferability to actual manufacturing conditions. This risk is exceptionally high for the named methods, which address the holistic development of process chains. The interfaces between the process steps themselves and the PEnv are often not systematically considered. Unconsidered factors such as particles, electrostatics, vibrations, temperature, and humidity affect process stability. Neglecting them can lead to variations in quality and costly errors in practice.

### 8.2. Practical Implications and Adaptation of Existing Methods

The example process chain presented in Section 7 illustrates the practical impact of external influencing factors. It demonstrates the quality variations and defects that may occur despite established standards if PEnv factors are not systematically considered. This example thus reinforces the previously identified need to integrate PEnv aspects already during process development. To reduce the risk of such planning failures, PEnv factors must be actively included in process planning. This consideration can be carried out in different ways using the following methods:Method by Vella et al.: Consideration of sub-processes and interfaces in a Process Pair Maturity Matrix (PPMM) to analyze how they influence each other.PDES: Simulation of processes based on known process data.µ-ProPlAn: Simulation of processes and process changes with the help of cause–effect networks and change propagation.Modified V-model by Watty and Binz: Consideration of PEnv via a checklist and additional methods.

Due to the fact that this consideration is not supported by the methods themselves but must be consciously extended by the user, a comprehensive consideration of the PEnv is not guaranteed in practice. This leads to a risk of errors due to incomplete consideration. As a result of the lack of systematic consideration in MST process development methods, the avoidance of process-specific failures due to the influence of PEnv [61,67,68,70,74,75,76] currently requires explicit analysis by internal or external experts in this field. This should be dealt with either through a targeted expansion of existing process development methods or through supporting methods. Methods that address operative process sequences can best be expanded due to their direct connection to the PEnv.

The most advanced integrations in this category are found in extended knowledge management systems such as PDES and µ-ProPlAn. As listed above, a PDES makes this possible with its linking of a knowledge database and simulation in XperiDesk software, in which the knowledge database can be expanded to include the necessary process parameters and simulations can be carried out. In this way, possible environmental influences can be simulated before real tests are carried out, for example, by testing the effects of different cleanroom levels in a virtual production environment. This predictive simulation makes it possible to identify and compensate for disruptive factors in the PEnv before they lead to quality problems in practice. In µ-ProPlan, cause–effect networks combined with change propagation offer a suitable way to consider changing PEnv influences and their potential impacts on process steps. This strength is particularly evident in process design and optimization when alternative process chains are compared in terms of their robustness to environmental conditions.

However, they require a continuously maintained database. Additionally, integrated analyses of the PEnv into existing extensive methods increase complexity and make their practical application more difficult. Adaptation to specific project requirements is also highly dependent on the user and requires in-depth expert knowledge as well as structured knowledge management. Due to the lack of a predefined structure, the results can vary significantly. If adapting existing methods is not appropriate, it is advisable to integrate supplementary methods based on the development requirements, according to Watty and Binz [57]. This procedure is only explicitly mentioned in the extended V-Model, but also makes sense for methods such as iMST and μPEP.

Currently, the most suitable tools for evaluating the PEnv and identifying possible negative influences are the generally applied PFMEA [146], or the expert questionnaire using the method of Vella et al. [22]. However, preparing such analyses or targeted surveys requires a profound knowledge of the critical parameters of the PEnv. Without sufficient expert knowledge, there is a risk in practice of a discrepancy between the expert knowledge in the field of MST and the required knowledge of the PEnv. This gap can lead to certain negative influencing factors or correlations being overlooked, which impairs the systematic analysis of the PFMEA or the creation of an expert questionnaire. Negative influencing factors that can be recognized during process development but cannot be completely excluded can be monitored via real-time monitoring of the subsequent process [155]. The cause–effect networks from µ-ProPlAn or a PFMEA are suitable for detecting such influencing factors.

### 8.3. Conclusion and Future Research

The mixed methods approach of SLR and modified GRADE analysis shows that many methods for process planning exist in MST. With the exception of the iMST method, no new methods have been published in recent years that cover the holistic development of process chains. In addition, none of the existing methods explicitly or adequately takes the PEnv into account. There is a need for research to develop systematic methods for the practical integration of the PEnv. A comprehensive method that integrates all aspects of process development, including the PEnv, would be impractical and would probably not become established in practice. Instead, a supporting method for explicitly analyzing the PEnv would be helpful. This could be designed as a modular evaluation framework in order to supplement existing methods in a targeted manner without unnecessarily increasing their complexity. The early integration of this perspective is essential, especially in the development of new processes and in existing processes that are subject to quality deviations due to external influencing factors. This promotes a more comprehensive awareness of potential external influences and enables systematic consideration of existing standards such as ISO 14644 or ISO 9001, which mandate the consideration of the PEnv [63]. Integrating the PEnv at an early stage prevents planning failures and accelerates development. It also enables a cost- and energy-efficient technical infrastructure design. An ergonomic and efficient PEnv also improves employee performance. This increase in efficiency is crucial in order to meet the challenges of progressive miniaturization and individualization in MST while ensuring the quality of production processes.

In the future, optimized manufacturing processes will be increasingly characterized by AI-based solutions that facilitate knowledge management and data acquisition, identify correlations, and ensure monitoring of production processes [156,157]. One example is embedding supervised learning in the methodological approach, as proposed by Civelek et al. [34,48]. Currently, these AI-based solutions are still under development, which limits their practical application and efficiency in production environments. Nevertheless, they offer great potential that should be further researched and implemented in the upcoming years.

## Figures and Tables

**Figure 1 micromachines-16-00606-f001:**
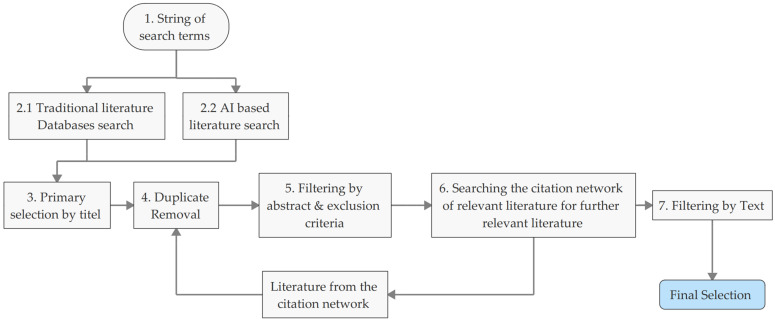
Methodology for a sequential literature search process.

**Figure 2 micromachines-16-00606-f002:**
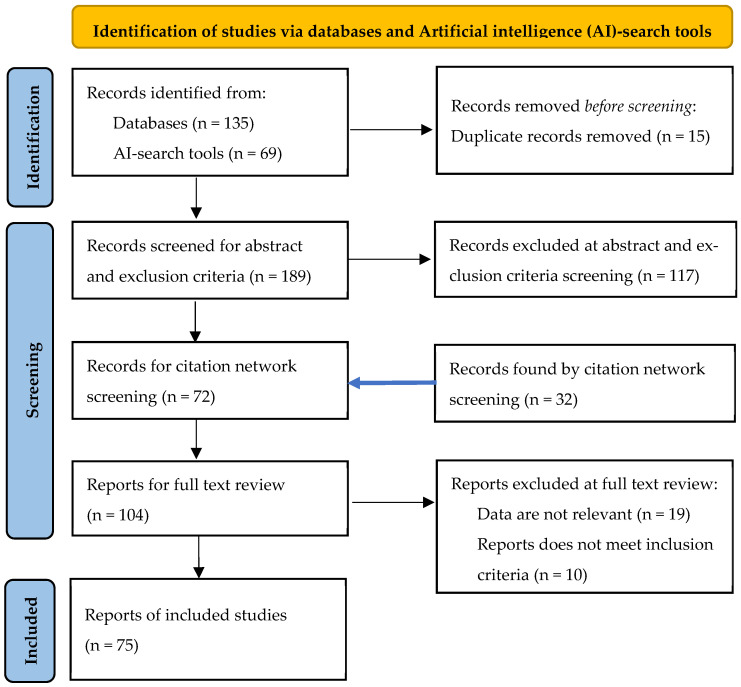
PRISMA flowchart of the literature researched in this SLR.

**Figure 3 micromachines-16-00606-f003:**
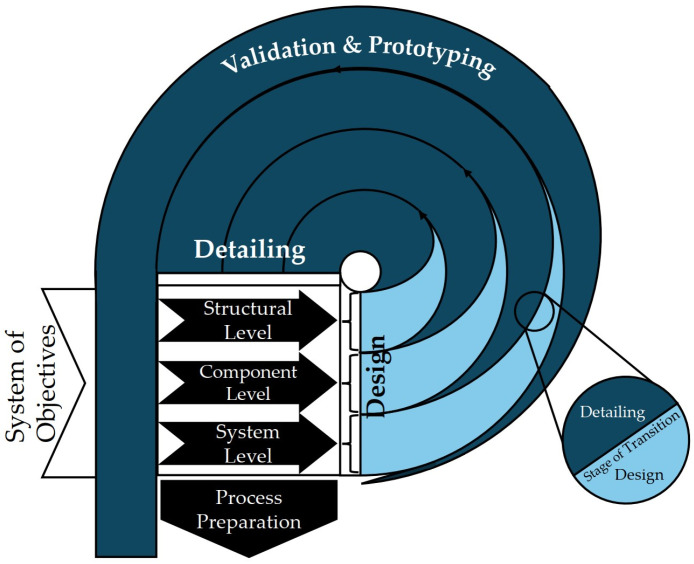
Sickle model for the design of tool-bound microsystems (graphic based on [104]).

**Figure 4 micromachines-16-00606-f004:**
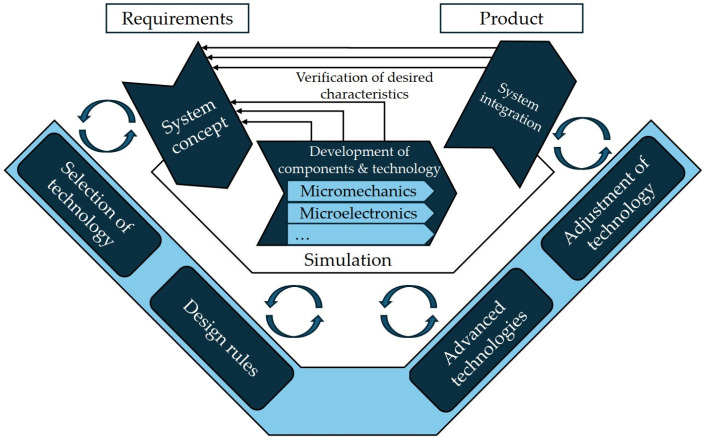
V-Model taking into account the process requirements (Graphic based on [36,151]).

**Figure 5 micromachines-16-00606-f005:**
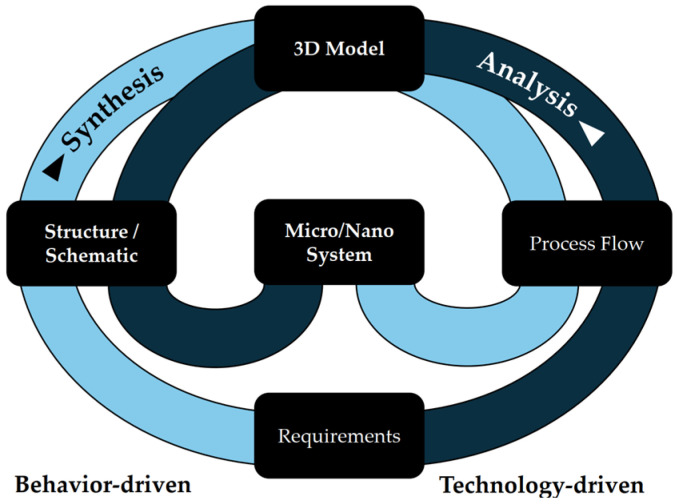
Pretzel model for MEMS design (graphic based on [24]).

**Figure 6 micromachines-16-00606-f006:**
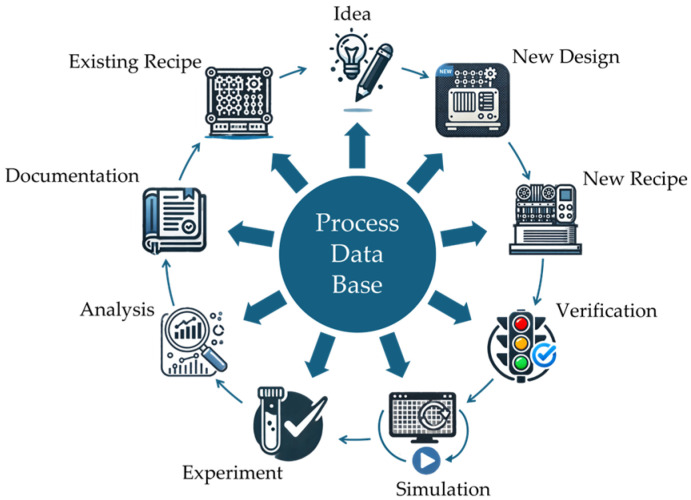
PDES-supported development cycle (graphic based on [24]).

**Figure 7 micromachines-16-00606-f007:**
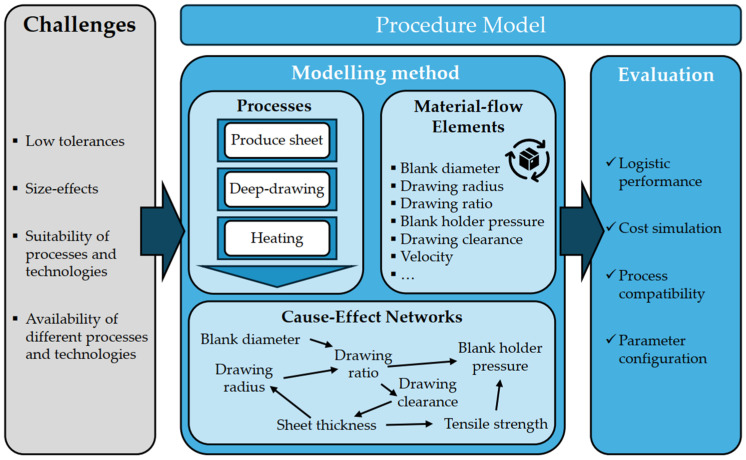
Components of the µ-ProPlAn framework (graphic based on [33,131]).

**Figure 8 micromachines-16-00606-f008:**
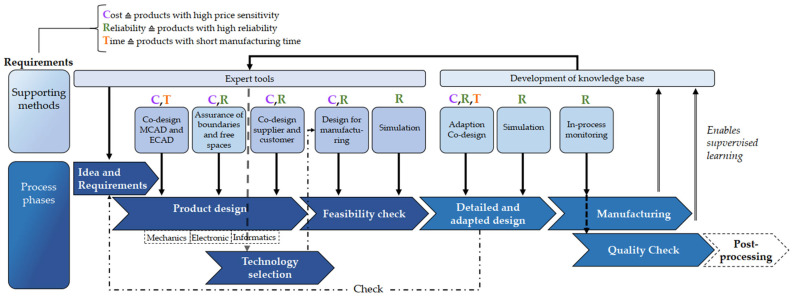
Product development process for individualized microsystems (graphic based on [34]).

**Figure 9 micromachines-16-00606-f009:**
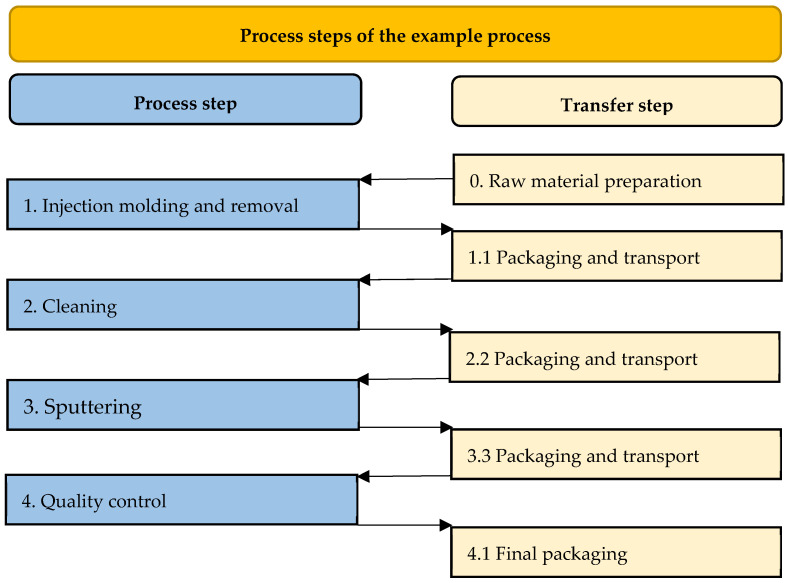
Process chain of the example process.

**Table 1 micromachines-16-00606-t001:** Overview of failure sources caused by the process environment.

Failure Sources	Failure Effects	Literary Sources
Temperature	Thermal expansion and tension of materials (measurement errors, cracks and delamination)	[61,67,68,70,71,72,73,74]
	Condensation on components	[70]
	Changes in chemical reactions	[70,72]
	Material aging and degradation	[67,70]
	Inaccuracies in the handling process of components due to changed adhesion	[61,67,75,76]
	Distortion of lithographic patterns	[70,72,74]
Humidity effects	Corrosion and oxidation	[68,77,78]
	Condensation defects	[72,79,80]
	Materials expansion caused by moisture	[68]
	Layer delamination and cracks	[74]
	Inaccuracies in the handling of components due to changed adhesion caused by capillary forces	[61,67,74,75,76,80,81,82]
	Electrostatic charge	[18,68,75,76]
Particles and contaminants	Corrosion caused by hygroscopic contaminants	[78,81]
	Uncontrolled doping of components	[68,83]
	Defects in lithography and etching steps	[18,72,74,79,83,84]
	Electrical short circuits	[60,61]
	Opacities of lithographic optics	[68]
Electrostatics	Inaccuracies in the handling process of components	[61,75,76,81,82]
	Sticking of parts or adhesion of particles due to electrical charge	[18,61,74,81]
Gases	Damage caused by outgassing components in the area around components or in transport containers	[18,79,85]
Vibrations and acoustic sources of vibration	Measurement error	[61,68]
	Inaccuracies in the handling process of components	[61,67,68,75]
Electromagnetic and radio frequency interference	Data loss or faulty signals	[68]
Human factors	Vibrations	[68]
	Incorrect cleaning	[68,74]
	Particulate impurities	[18,68]

**Table 2 micromachines-16-00606-t002:** Overview of the keywords used.

Subject Area	Keywords
Development	process-/product development
	process-/product engineering
	process planning
	process optimisation
	process design
Microsystems technology	microtechnology
	nanotechnology
	microsystem technology
	microengineering
	micro-electromechanical system
	MEMS
Process environment	technical cleanliness
	cleanroom
	contamination control
	process environment
	fabrication environment
Methodical approach	method
	technique
	strategy
	approach

**Table 3 micromachines-16-00606-t003:** Inclusion and exclusion criteria for selecting the literature were found.

Inclusion Criteria	Exclusion Criteria
Product development methods for microsystems technology, in which process planning is also considered	General planning methods that are not focused on microsystems technology
Development methods for parts of process planning in microsystems technology	Methods for assessing the impact of processes on nature and the climate
Development methods for the whole process planning in microsystems technology	Methodological approaches for technical processes

**Table 4 micromachines-16-00606-t004:** List of literature databases and AI tools searched.

Conventional Literature Databases and Publishers	AI Tools
arXiv	Chat GPT
Google Scholar	Connected Papers
IEEE	Elicit
Jstor	Litmaps
MDPI	Paperdigest
PubMed	Perplexity
ResearchGate	Scholar AI
ScienceDirect	SciSpace
Scopus	
SpringerLink	
Web of Science	
Wiley	

**Table 5 micromachines-16-00606-t005:** Overview of methods for process planning in MST.

Category		Software-Based Methods 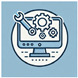	Framework-Based Methods* 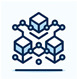 *
Production-oriented product development (PPD)	* 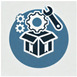 *	[15,40,46,52,103,104,105,106,107]	[8,10,11,27,34,39,46,47,48,56,57,66,104,107,108,109,110,111]
Development/planning of process chains (DPP)	* 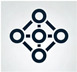 *	[9,13,14,22,24,43,50,51,52,53,112,113,114,115,116,117,118,119,120,121,122,123,124]	[8,9,11,13,15,23,24,33,34,36,42,44,48,50,53,58,66,114,116,118,125,126,127,128,129,130,131,132]
Selection of manufacturing processes (SMP)	* 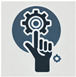 *	[9,13,15,24,34,43,52,53,112,113,115,116,117,118,123,133,134]	[9,11,23,36,46,48,56,58,66,125,126,127,129]
Material selection (MS)	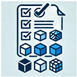	[9,13,24,43,52,53,113,116,117]	[9,23,34,46,133,135,136,137,138,139,140,141,142,143]
Material flow planning (MFP)	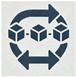	[13,24,43,52,53,116,117,118,144]	[23,34,44,126,129]
Possibility to include the process environment	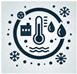	[13,22,53,124,133,145]	[15,39,57,146]

**Table 6 micromachines-16-00606-t006:** Overview of selected process development methods in MST.

Method	Application in the Development Process	PEnv Explicit?	Data-Based?	Model-Based?	Main Advantage
μPEP	Early product development	No	No	No	Combining design rules and requirements for MST manufacturing processes
μ-ProPlAn	Process design and evaluation	Yes (implicit)	Yes	Yes	Model-based analysis of interdependencies
V-model (Watty & Binz)	Analysis of early manufacturing requirements	Yes (checklist)	No	No	Systematic consideration of PEnv
iMST	Customized system configuration	No	No	No	High degree of adaptability to individual projects
PDES	Continuous process development	Yes (implicit)	Yes	Simulation	Digital management and simulation of process variations

**Table 7 micromachines-16-00606-t007:** Evaluation criteria of the modified GRADE analysis.

Evaluation Criteria		Explanation of the Evaluation Criterion
1	Risk of bias	This evaluation criterion checks if the methods are potentially susceptible to systematic failures if the process environment is not considered.
2	Indirectness of evidence	This assesses if the category of methods is suitable for the specific application of considering the process environment. It analyzes the extent to which the methods are appropriate for integrating environmental influences.
3	Imprecision of the results	This criterion assesses the accuracy of results related to the process environment. Precise data from process planning is required for planning and analyzing the process environment. Methods that provide inaccurate or incomplete data are assessed more critically.
4	Adaptability for considering the process environment (new category)	This criterion evaluates whether methods can be flexibly adapted or extended to consider specific requirements of the process environment.
5	Publication bias	This criterion evaluates whether methods for planning and assessing a suitable process environment are underrepresented in the literature and whether the inclusion of the process environment is even meaningful in the respective category.

**Table 8 micromachines-16-00606-t008:** Quality levels of the evidence.

Quality Level	Explanation of the Quality Level
High	The evidence is of high quality, and there is great confidence that the method provides a reliable result in the context of the process environment.
Moderate	The evidence is solid but has some uncertainties regarding the process environment that could slightly reduce confidence.
Low	The evidence is limited, or there are significant uncertainties regarding the process environment that could substantially impair confidence.
Very low	There is little confidence in the evidence regarding the process environment, as it may be significantly biased or incomplete.

**Table 9 micromachines-16-00606-t009:** Summary of findings.

Method Categories	Risk of Bias	Indirectness of Evidence	Imprecision of the Results	Adaptability for Considering the Process Environment	Publication Bias	Overall Evaluation
Production- oriented product development	Low	Low	Very low	Low	Low	Low
Development/planning of process chains	Low	Low	Low	Moderate	Very low	Low
Selection of manufacturing processes	Moderate	Very low	Very low	Low	Moderate	Very low
Material selection	Moderate	Very low	Moderate	Low	Moderate	Low
Material flow planning	Moderate	Very low	Very low	Moderate	High	Very low

**Table 10 micromachines-16-00606-t010:** Possible process environment influences and effects of the example processes.

Process Step	Possible Process Environment Influences	Potential Component Defects	Method Category
0	Excessive humidity	Defectively manufactured component due to residual moisture in the raw material	DPP, MS, MFP
0	Contamination that gets onto the raw material during handling—the resulting electrostatic charge increases the effect	Defectively manufactured components due to contamination in the raw material	DPP, MS, MFP
1	Particles and impurities due to inadequate cleanroom class or contamination generated by the process	Defective microstructures due to impurities in the component	PPD, DPP, SMP
1	The hot thermals of the injection molding tool create an upward airflow	Particles from the floor or the machine bed contaminate the component	DPP, SMP
1, 2, 3, 4	Process influences from workers operating the machine	Contamination due to improper cleaning or manual handling and packaging of the component	PPD, DPP
1, 1.1, 2.1, 3.1	Electrostatic charging of the component due to the injection molding process, the plastic transport container, or humidity	Particles adhering to the component due to electrostatic charge	PPD, DPP, SMP
2.1, 3	Particles and contaminants due to inadequate cleanroom classification after cleaning	Enclosed contaminants after sputtering	DPP, MFP
3	Condensation formation due to temperature differences	Holes in the surface coating	DPP, MFP
4	Vibrations caused by nearby external processes or personnel	Measurement errors in quality control	PPD, DPP
1.1, 2.1, 3.1, 4.1	Contamination in packaging due to improper cleaning or incorrect storage	Contamination of the component due to transport packaging	DPP, MFP

## Data Availability

The original contributions presented in this study are included in the article. Further inquiries can be directed to the corresponding author.

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
