# Peer review of "Process Development Methods in Microtechnology and the Associated Process Environment"

_micromachines, 2025, doi:10.3390/mi16060606_

Round 1
Reviewer 1 Report
Comments and Suggestions for Authors
This article is devoted to the analysis of technical literature related to the development of processes in microtechnologies. As an important result of this article, I will point out the following fact. In each of sections акщь 3 to 8, the following subsections are included with some deviations: a) Development planning of technological chains, b) Choice of production processes, c) Choice of material, d) Planning the flow of materials. Moreover, in section 2, this principle of dividing a section into subsections of the description is justified in detail. It seems to me that it would be desirable for the reader to begin by giving a hierarchical classification of the article into sections and their subsections (something like an Outline) in order to make the presentation of rather difficult material on microtechnology more understandable. I would also like to describe in more detail in the Introduction the selection of various sections (3-8) of the article. In general, the paper is very interesting and informative. As a reviewer, I was just trying to characterize the article not from the point of view of the technology itself, but from the point of view of its information component and the ease of use of the very useful review offered by the authors to various readers interested in microtechnology. In general, the paper is very thorough, informative and can certainly be published in the Micromachines journal.
Author Response
Thank you for your time and the review. Please find a detailed description of the improvements made in the attached Word file.

Reviewer 2 Report
Comments and Suggestions for Authors
The manuscript tried to present a review of process development and process environment related influences in microtechnology. However, this review lacks novelty and research significance. Much of content is on literature review methods , summarization of categorized literatures, and general methodologies, the delivery is not significant or in-depth.
Author Response

(The authors gave the same response as above.)

Reviewer 3 Report
Comments and Suggestions for Authors
This manuscript focuses on the correlation between microsystem technology (MST) and microelectromechanical systems (MEMS) process development and process environment (PEnv). The topic is highly cutting-edge and important, and has significant implications for improving manufacturing efficiency and product quality in this field.
However, when explaining the application cases of specific process development methods, the details of some cases are not rich enough, resulting in insufficient presentation of the advantages and limitations of the methods in practical applications. After modification, it can be considered for acceptance. Such as:
- When introducing the application of the μ - ProPalAn method, there is a lack of specific data and response measures on how to use this method to deal with sudden environmental changes in actual production.
- When analyzing the consideration of PEnv by existing methods, the focus is mainly on the shortcomings of existing methods, and there is relatively little discussion on the relative advantages of different methods considering PEnv in specific scenarios.
- When introducing different process development methods, the comparison and connection between each method are not natural enough, making it difficult for readers to quickly grasp the differences and connections between different methods.
Author Response

(The authors gave the same response as above.)
